

# A feature-enhanced knowledge graph neural network for machine learning method recommendation

Xin Zhang and Junjie Guo

School of Artificial Intelligence and Big data, Hefei University, Hefei, China

## ABSTRACT

Large amounts of machine learning methods with condensed names bring great challenges for researchers to select a suitable approach for a target dataset in the area of academic research. Although the graph neural networks based on the knowledge graph have been proven helpful in recommending a machine learning method for a given dataset, the issues of inadequate entity representation and over-smoothing of embeddings still need to be addressed. This article proposes a recommendation framework that integrates the feature-enhanced graph neural network and an anti-smoothing aggregation network. In the proposed framework, in addition to utilizing the textual description information of the target entities, each node is enhanced through its neighborhood information before participating in the higher-order propagation process. In addition, an anti-smoothing aggregation network is designed to reduce the influence of central nodes in each information aggregation by an exponential decay function. Extensive experiments on the public dataset demonstrate that the proposed approach exhibits substantial advantages over the strong baselines in recommendation tasks.

## INTRODUCTION

In the last decades, the field of machine learning has witnessed a surge in the emergence of numerous machine learning methods and datasets. The overwhelming amounts of machine learning methods have resulted in severe information overload. Thus, it becomes difficult for researchers to select the appropriate machine learning approach for a given dataset in scientific research. Therefore, some efforts must be taken to automatically recommend the most suitable machine learning method for a target dataset.

Actually, recommender systems (*Guo et al., 2020*; *Wang et al., 2024*) provide an effective solution to the information overload problem. In recent years, knowledge graph-based recommender systems have achieved great success in the fields of news recommendation (*Wang et al., 2018*; *Qiu, Hu & Wu, 2022*), movie recommendation (*Cheng et al., 2020*), and commodity recommendation (*Ma et al., 2019*; *Li et al., 2023*). Inspired by this, researchers have attempted to recommend the most suitable machine learning method for a target dataset by incorporating knowledge graph. For example, a description-enhanced approach (*Cao et al., 2021*) was devised by leveraging the knowledge graph and descriptive

Corresponding author
Xin Zhang, zhangxin@hfuu.edu.cn

information as the auxiliary information to improve the quality and efficiency of picking up the machine learning method for a target dataset. To further improve the performance of machine learning method recommendation, cross-modal knowledge graph contrastive learning (*Cao et al., 2022*) maximized the consistency of graph structure modality and description text modality by introducing contrastive learning. However, the graph neural network, a core component in the knowledge graph-based recommendation method, needs more support in fully utilizing the node feature information to further improve the recommendation efficiency in machine learning applications.

Figure 1 in *Cao et al. (2021)* shows that the rich connections in the machine learning knowledge graph can be utilized to recommend the appropriate method for the target dataset. For example, the *MovieLens-20M* dataset is taken as a seed node in this figure. The first-order neighbors of *MovieLens-20M* include its relevant tasks (*e.g.*, *Recommender Systems*) and its adopted methods (*e.g.*, Knowledge Graph Convolutional Network (KGCN)). The first-order neighbors provide information about which tasks *MovieLens-20M* is suitable for and which methods it is used by. The second layer captures entities that overlap with interactive items such as *authors*, *papers*, *Bing-News*, and *Last-FM*. These entities are connected to the target entity through other entities and have some relationship and similarity with the target entity. Reasoning through the connectivity between entities helps to generate correct recommendations, which is the key to knowledge graph-based recommend systems (*Wang et al., 2019b*). However, in most knowledge graph-based recommendation methods, each node is only represented by its initial representation, and informative messages carried by its neighbors are ignored. Actually, more hidden messages are carried by the neighbors of each node. For example, when the aggregation process involves the node *Last-FM*, its initial representation, which only contains its name, is insufficient. The node representation should convey other relevant messages, such as information about the method *MKR* applied on the dataset *Last-FM*. Furthermore, as the number of layers increases, the embedding of each entity accumulates an expanding amount of information. This accumulation tends to obscure the fine-grained details of the node itself, making it difficult to distinguish from the remaining nodes, which is the embedding over-smoothing (*Li, Han & Wu, 2018*) issue.

Inspired by these observations, the article proposes the feature-enhanced knowledge graph neural networks for machine learning method recommendation (FEGNN). The FEGNN method enhances the structural information of the knowledge graph by enriching entity representation and alleviating the effects of over-smoothing, thereby improving the recommendation effectiveness.

The contributions of this article are listed as follows.

• Although the representation of each target entity can be enhanced by the textual description information in the machine learning knowledge graph, a closer relationship between the first-order neighbors of a node and the node itself is captured. The representation for each node is enhanced through its neighborhood information before participating in the higher-order propagation of the target entity process.

• A novel framework that integrates the feature-enhanced graph neural network and an anti-smoothing aggregation network is devised. Over-smoothing issue may be caused by

the multiple stacks of several graph neural network layers. Different from DEKR, to prevent the representation of the target entity from being over-smoothing, an exponential decay function is incorporated into FEGNN to control the influence of the original central node in the information transfer process.

- Extensive experiments are conducted on the public real datasets. The experimental results demonstrate that the proposed FEGNN method improves Top-K recommendation and click-through-rate prediction over the strong baselines.

## RELATED WORK

This section introduces graph neural networks, knowledge graphs and knowledge graph-based recommendation methods.

### Graph neural networks

Graph neural networks (GNN) are widely applied in recommendation systems by utilizing auxiliary graph-structured information. GNN offers a unified framework to manage the substantial data in recommender systems and explicitly encode critical collaborative signals to enhance the representations of users and items. Numerous GNN variants have been proposed in recent years. For instance, dynamic representation learning *via* recurrent graph neural networks (*Zhang et al., 2023*) designed a one-stage model that integrates a recurrent neural network into GNN to generate compact representations. Gated recursion-based graph neural network (*Ge, Zhao & Zhao, 2022*) proposed a gated recursive algorithm in order to address the node aggregation issue and extract deep dependency features among nodes. Multi-relational graph attention networks (*Li et al., 2022b*) captured more complex semantic relationships between entities through an attention mechanism and embedding edge relationship types. Although GNN-based recommendation systems have enhanced the effectiveness of recommendation, it is necessary to exploit more comprehensive graph structures and semantic relationships between entities. Thus, researchers attempt to apply the graph neural network in knowledge graphs to further improve the recommendation performance.

### Knowledge graph-based recommender systems

A knowledge graph is a graphical database that is specifically crafted to organize, represent, and store structured knowledge (*Li, Qu & Wang, 2023*). Its fundamental concept was visually representing real-world information and concepts with a triple *<entity, relationship, entity>*. A knowledge graph is utilized in various domains, such as question answering, text classification and recommendation systems (*Qiu et al., 2020*; *Zhu et al., 2023*; *Cui et al., 2022b*; *Zhang et al., 2022b*). Recommendation methods based on knowledge graphs can be divided into three categories: embedding-based, path-based and propagation-based methods.

**Embedding-based methods.** The embedding-based methods leveraged knowledge graph information to enhance the representation of each entity. The entities and relationships in the knowledge graph were encoded as low-dimensional vectors in the embedding-based methods to preserve the inherent graph structure. For instance,

personalized recommendation system based on knowledge embedding and historical behavior (*Hui et al., 2022*) used self-attention to determine user preferences and integrated knowledge graph embeddings with user behavior. The deep interest network based on knowledge graph embedding (*Zhang et al., 2022a*) addressed the limitations of static interaction matrices by combining gated recurrent units with an attention mechanism. A knowledge graph-based approach for visualization recommendation (*Li et al., 2022a*) enhanced the performance of TransE by replacing evidently incorrect triples with self-adversarial negative sampling. Despite the simplicity and flexibility, these approaches overlooked the complex semantic relationships between entities in the knowledge graph, which leads to the fact that the entity expression in the knowledge graph was monotonous.

**Path-based methods.** The path-based approach enhanced recommendations by identifying connectivity similarities among paths from users to items. This was achieved by constructing a user-item graph and leveraging entity connectivity to enhance recommendation effectiveness while preserving interpretability. For instance, reinforced sequential learning with gated recurrent unit (*Cui et al., 2022a*) combined reinforcement path reasoning network components and gated recurrent units to enhance path reasoning capabilities. Reinforcement learning framework for multi-level recommendation reasoning (*Wang et al., 2022*) addressed local optimization issues by incorporating abstract markov decision processes and developed a multi-layer path extraction algorithm to improve model performance. Path language modeling recommendation (*Geng et al., 2022*) predicted new paths based on higher joint path probability allocation scores, effectively extending the reachability of items that traditional methods cannot achieve. However, valuable information may be lost by separating complex user-item connectivity into discrete linear paths in the path-based recommendation methods.

**Propagation-based methods.** The propagation-based recommendation methods optimized the utilization of knowledge graph information by integrating connectivity information and semantic representations of entities and relationships. These methods utilized embedding propagation, which aggregated the embeddings of multi-hop neighbor nodes in the knowledge graph, to enhance entity representation. These methods facilitated the anticipation of user preferences by comprehensively representing users and items. For instance, knowledge graph convolutional networks (*Wang et al., 2019b*) and knowledge graph convolutional networks with label smoothness (*Wang et al., 2019c*) leveraged graph convolutional networks to compute neighborhood-propagated embeddings of items in knowledge graphs. Description enhanced knowledge graph recommendation (*Cao et al., 2021*) overcame the limitations of lacking textual descriptive information by combining knowledge graph-based and text-based approaches. Cross-modal knowledge graph contrastive learning (*Cao et al., 2022*) learned node representations by considering descriptive attributes and structural connections as two modalities and maximizing the consistency between them. Knowledge-adaptive contrastive learning (*Wang et al., 2023*) learned the generated user-item interaction view and knowledge graph view by introducing contrastive learning. These techniques took the entire knowledge graph as input, obtaining the embedding for each entity by aggregating information from all its neighbors. However, the recommendation performance may be hurt by the lack of

constraints between the target entity and its neighbors since the embedding learning process inevitably introduces noise (less informative neighbors) and involves substantial computation (a large number of neighbors).

The influence of direct neighbors during higher-order propagation is ignored in these propagation-based methods. In this article, the features of nodes involved in the high-order propagation process of the target entity are enhanced through their first-order neighbor information. An anti-smoothing aggregation network is designed to reduce the impact of over-smoothing.

## PROBLEM FORMULATION

This section outlines the formulation of the FEGNN problem. The article defines the machine learning datasets as $D = \{d_1, d_2, \ldots, d_M\}$ and the machine learning methods as $M = \{m_1, m_2, \ldots, m_N\}$. The dataset-method interaction matrix $Y = \{y_{dm} | d \in D, m \in M\}$ can be represented as follows:

$$y_{dm} = \begin{cases} 1, & \text{if } (d, m) \text{ interaction record exists}, \\ 0, & \text{otherwise}. \end{cases} \tag{1}$$

The knowledge graph $G$ consists of numerous triples, expressed as $G = \{(h, r, t) | h, t \in \varepsilon, r \in R\}$, where $h$ is the head entity, $t$ is the tail entity, and $r$ is the relationship between them. $\varepsilon$ and $R$ represent the sets of entities and relationships in the knowledge graph. For instance, the triple (*Last-FM, used, KGCN*) indicates that the *Last-FM* dataset is used by the *KGCN* method, and the *KGCN* method employs the *Last-FM* dataset conversely. This article uses the rich auxiliary information in the knowledge graph by linking the target entities to other machine learning knowledge graph entities. Two types of nodes in the knowledge graph are defined. Specifically, nodes with descriptive information are defined as descriptive nodes $e^t$, and the remaining entities are general nodes $\{e_1^g, e_2^g, \ldots, e_n^g\}$. Descriptive nodes are excluded from the outward extension process.

The representations of nodes involved in target entity propagation are enhanced by aggregating their direct neighbor information in the knowledge graph. These feature-enhanced nodes are denoted as $G = \{g_1, g_2, \ldots, g_i\}$. The ultimate representation of the target entity is derived from these feature-enhanced nodes.

Furthermore, the textual descriptive information is incorporated as the description-enhanced features for both dataset and method entities in the machine learning knowledge graph. Compared with the traditional knowledge graph, the fundamental concepts and operational attributes can be obtained through these textual descriptive messages. Specifically, $t_d \in T$ and $t_m \in T$ represent the descriptive information of the dataset and the method, respectively. $T$ is a set of descriptive documents.

The machine learning recommendation task involves predicting the interaction probability between a given dataset and the methods by leveraging the auxiliary information from the machine learning knowledge graph. This article aims to predict the interaction probability between datasets and methods for applications of lacking interaction records. This can be expressed as $\hat{y}_{dm} = \mathcal{F}(d, m, \Theta)$, where $\hat{y}_{dm}$ denotes

the interaction probability, and $\Theta$ represents a parameter set within the predictor function $\mathcal{F}$.

## METHODS

### Overall framework

The proposed FEGNN approach consists of two main components. The first component is a graph neural network that integrates the feature-enhanced graph neural network and an anti-smoothing aggregation network by exploring the high-order connectivity of entities in the machine learning knowledge graph. The other component is a deep text-based collaborative filtering network that captures linear and nonlinear textual content interactions. Finally, the outcomes of the two components are integrated to predict the interaction probability of a particular target entity. The proposed FEGNN approach is illustrated in Fig. 1.

All of the parts mentioned below contribute to the novel component, namely "feature-enhanced knowledge graph neural network". The first part of FEGNN is "Linking and Propagating". It can be easily observed that the proposed method takes the target dataset and method as inputs. In the "Linking and Propagating" part, the target dataset and candidate machine learning method are initially linked to the corresponding entity nodes in the machine learning knowledge graph $G$. The linked entities $e_d$ and $e_m$ serve as the seed nodes. Different from the propagation process in DEKR, the features of general nodes involved in the propagation process for the target entity are enhanced by their first-order neighborhood information, as shown in yellow. The article defines these feature-enhanced nodes as $\{g_1, g_2, \ldots, g_i\}$. Then, the target node extends progressively to higher-order neighbor nodes through these feature-enhanced ones. This article designs an anti-smoothing aggregation network to acquire entity representations to predict the probability $\hat{y}_1$.

Similar to DEKR, the article uses the text-based deep collaborative network component to capture interaction information in the textual descriptive information and predict the probability $\hat{y}_2$. Both components determine the ultimate interaction probability of the target dataset and method.

### Feature-enhanced knowledge graph neural network

**Feature enhancement.** On the basis of the knowledge graph, graph neural networks provide a method of further exploiting the rich connections between entities. Considering the close relationship between a node and its direct neighbors, these neighbors may convey informative messages. For instance, the propagated node represents *a machine learning method*, and its embedding information in the traditional knowledge graph method may only include the name, which is insufficient. More messages in the knowledge graph method such as its associated task, datasets and authors, are needed to explain *a machine learning method* better. These additional messages can be obtained from its direct neighbors in the knowledge graph. Therefore, the article actively enhances the features of these general nodes $\{e_1, e_2, \ldots, e_i\}$ involved in the propagation process of the target entity as feature-enhanced nodes $\{g_1, g_2, \ldots, g_i\}$ by aggregating information from itself and its

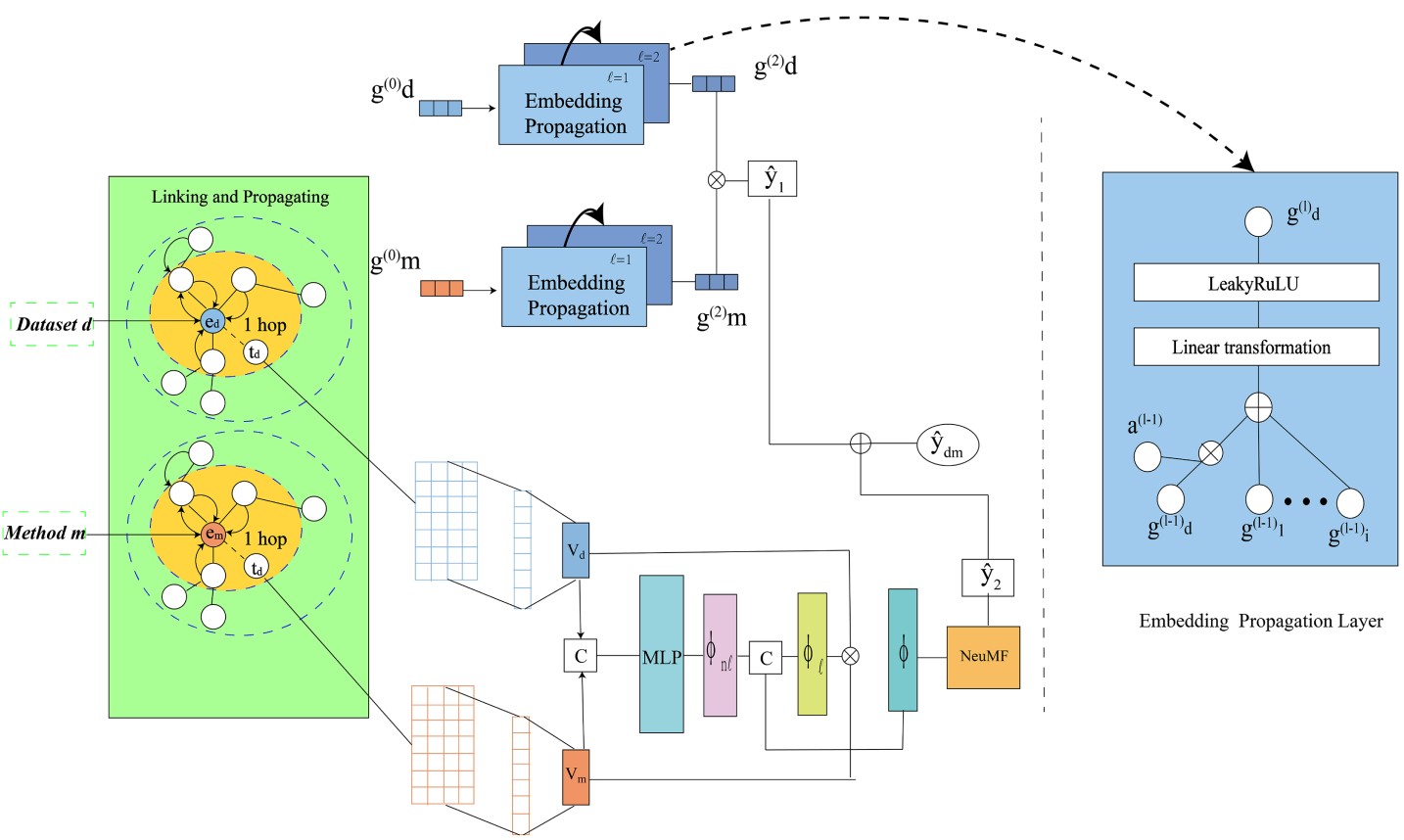

**Figure 1 Framework of feature-enhanced knowledge graph neural networks for machine learning method recommendation (FEGNN).**

direct neighbors. Then, the target entity is propagated over these feature-enhanced nodes to obtain a more comprehensive higher-order representation.

Nevertheless, the significance of the neighbors for a target entity should not be uniformly assessed. For example, given a specific dataset, a machine learning method experimented on the dataset theoretically provides more impact than the author with which the dataset has interacted. This distinction arises because the methods using the same dataset typically address similar tasks, while authors may be involved in diverse fields. Therefore, it is crucial to distinguish the importance of the neighbors. In this article, the following formula is designed to calculate the attention scores for different relations ($r$) of node ($e$):

$$\tilde{\pi}_{r,e_t} = \frac{exp\left(\pi_{r,e_t}\right)}{\sum_{e_t \in N(e)} exp\left(\pi_{r,e_t}\right)}. \tag{2}$$

where $e_t$ is the neighbor of node $e$, $\tilde{\pi}_{r,e_t}$ represents the relevant weight of neighbors, and the higher weights indicate the necessary to transmit more information for the target node. As each node has a varying number of neighbors, a fixed sampling strategy is employed to enhance the training efficiency.

A weighted summation approach is used to calculate the representation of neighborhood information. This approach utilizes correlation weights derived from Eq. (2) denoted as $Nei$.

$$e_{Nei} = \sum_{e_t \in N(e)} \tilde{\pi}_{r,e_t} e_t. \tag{3}$$

Thus, more informative messages are exploited by means of integrating the neighbor information and its own information. The calculation formula is as follows:

$$g = LeakyReLU\left( W_0 \left( e^{(0)} + e_{Nei}^{(0)} \right) + b_0 \right). \tag{4}$$

Here, LeakyRelu is adopted as the activation function, which can handle both positive and negative signals. $W_o$ and $b_0$ are the learnable parameters.

Consequently, high-order representations of entities are derived based on these feature-enhanced nodes $\{g_1, g_2, \ldots, g_i\}$.

## Anti-smoothing aggregation networks

**One-hop Anti-smoothing Propagation.** A single propagation layer is taken as an example to better explain the mechanism of the anti-smoothing aggregation network. A specific dataset-method pair is treated as the seed node in this illustrated example. Once these nodes are mapped to the corresponding nodes $(e_d - e_m)$ in the knowledge graph, connections can be established with other entities (first-order neighbors) through various relationships. Distinguishing the contributions of different relations $(r)$ for the feature-enhanced node $(g)$ is essential. Therefore, the following formula is used to calculate the contribution weights:

$$\tilde{\pi}_{r,g}^d = \frac{exp\left(\pi_{r,g}^d\right)}{\sum_{g \in N(m)} exp\left(\pi_{r,g}^d\right)}. \tag{5}$$

$$\tilde{\pi}_{r,g}^m = \frac{exp\left(\pi_{r,g}^m\right)}{\sum_{g \in N(d)} exp\left(\pi_{r,g}^m\right)} \tag{6}$$

Here, $\tilde{\pi}_{r,g}^d$ and $\tilde{\pi}_{r,g}^m$ denote the correlation weights calculated by $N(m)$ and $N(d)$ with reference to $g_d$ and $g_m$, respectively.

Then, a weighted summation is adopted to indicate the aggregated neighborhood information using the formula below:

$$g_{N(d)} = \sum_{g \in N(d)} \tilde{\pi}_{r,g}^d g \tag{7}$$

$$g_{N(m)} = \sum_{g \in N(m)} \tilde{\pi}_{r,g}^m g. \tag{8}$$

In the following step, the node's self-information is updated by merging the information of its neighbors during the propagation process to accumulate more enriched messages

with each convolutional layer. At the same time, node features are augmented to provide richer information for the FEGNN.

However, these may exacerbate the issue of over-smoothing. Considering these factors, an anti-smooth aggregation network is proposed for the aggregation process. This structure can systematically reduce the effect of self-information during each aggregation. The goal is to prevent an undue reliance on progressively enriched self-information to alleviate the problem of over-smoothing. The calculation formula for the anti-smoothing structure is shown below.

$$a_l = \alpha * e^{-(l+1)} \tag{9}$$

where $l$ is denoted as the current number of propagation layers, and $\alpha$ is a decay parameter that adjusts smoothing. After aggregating neighborhood information, the final representation in one-hop propagation for a given entity is shown as follows.

$$g_d^{(1)} = LeakyReLU\left(W_0\left(a_0 * g_d^{(0)} + g_{N(d)}^{(0)}\right) + b_0\right) \tag{10}$$

$$g_m^{(1)} = LeakyReLU\left(W_0\left(a_0 * g_m^{(0)} + g_{N(m)}^{(0)}\right) + b_0\right) \tag{11}$$

where $W_0$ and $b_0$ are the learnable parameters, LeakyReLU is adopted as the activation function. $g_d^{(0)}$ and $g_{N(d)}^{(0)}$ denote the original representations of the node and its neighbors, respectively.

**Higher-order propagation.** The ultimate representation of the entity is acquired by multiple layers of propagation to capture more complex connection details and information from distant neighbors. The computational formula is expressed as follows.

$$g_d^{(l)} = LeakyReLU\left(W_{l-1}\left(a_{l-1} * g_d^{(l-1)} + g_{N(d)}^{(l-1)}\right) + b_{l-1}\right) \tag{12}$$

$$g_m^{(l)} = LeakyReLU\left(W_{l-1}\left(a_{l-1} * g_m^{(l-1)} + g_{N(m)}^{(l-1)}\right) + b_{l-1}\right). \tag{13}$$

In this equation, $W_{l-1}$ represents the trainable weight matrix, and LeakyReLU is configured as the activation function. $g^{(l-1)}$ denotes the entity representation derived from the preceding aggregation layer, and the higher-order representation for a target entity is obtained by multiple high-order propagations.

**Prediction layer.** After multiple hops of aggregation, the final representations for the target dataset and machine learning method, namely $g_d^{(l)}, g_m^{(l)}$, are derived by incorporating information from its $l$-hop neighbors. The interaction probability between the target dataset and the machine learning method is predicted by leveraging the graph neural network, as shown below.

$$\hat{y}_1 = \sigma\left(\left(g_d^{(l)}\right)^T g_m^{(l)}\right) \tag{14}$$

where $\sigma$ is the sigmoid activation function.

## Text-based collaborative filtering

The graph neural network captures the structure information by utilizing auxiliary knowledge graph data. However, relying solely on the graph structure poses a challenge due to the brevity of the illustration of machine learning methods and datasets, which leads to factual inconsistencies. For instance, the path in (*MovieLens-20M, dataset. method, DeepFM*), (*DeepFM, method. dataset, Bing News*), (*Bing News, dataset. method, DKN*) might incorrectly suggest *MovieLens-20M* is suitable for *DKN*. Actually, *DKN* is a text-based recommendation method, while *MovieLens-20M* is a film dataset without effective textual information.

This article uses a content-based deep collaborative filtering network to predict the interaction probability for the target dataset and method. Note that the description nodes of core entities are not engaged in the propagation to higher-order neighbors. For the textual description related to the target entity, the $n$ words of the original description are represented as $t = w_{1:n} = [w_1, w_2, \ldots, w_n]$. The matrix $s_{1:n} \in R^{p \times n}$ signifies the embedding matrix of the sentence. Pretrained words to vectors are produced on GloVe (*Pennington, Socher & Manning, 2014*), the initial embedding representation for each word in which global statistical information and local contextual features are contained. The sentence representation $s_t \in R^p$ is derived by aggregating the average of the word representations.

For the descriptive information $e^t_{(d)}$ and $e^t_{(m)}$ associated with a given dataset-method pair $g_{(d)}$-$g_{(m)}$, the low-dimensional vectors $v_d$ and $v_m$ are obtained through matrix transformation. This process can be expressed as follows.

$$v_d = W_d s_d \tag{15}$$
$$v_m = W_m s_m. \tag{16}$$

To further explore the interaction between the dataset and the machine learning method, a neural collaborative filtering framework (*He et al., 2017*) is utilized in this article. Generalized matrix factorization captures linear interactions in the descriptive information for the dataset and method. Simultaneously, a multi-layer perceptron layer captures nonlinear interactions in the descriptive information. A concept similar to matrix factorization is employed to extract linear interaction between the dataset and method description features.

$$\phi_l = v_d \odot v_m \tag{17}$$
$$\hat{y}_l = \sigma\left(G^T \phi_l\right). \tag{18}$$

Here, $\odot$ denotes the element-wise product. $G^T$ denotes the learnable weight matrix. A sigmoid activation function is adopted as $\sigma$.

Meanwhile, the two feature vectors $v_d$ and $v_m$ are concatenated to capture nonlinear interaction features through a multi-layer perceptron. The formulation is described as follows.

$$h_o = v_d \parallel v_m \tag{19}$$

$$\phi_{nl} = h_n = \sigma\left(W_n^T h_{n-1} + b_n\right) \tag{20}$$

$$\hat{y}_{nl} = \sigma\left(M^T \phi_{nl}\right). \tag{21}$$

Here $\|$ donates the concatenation operation of two vectors, and $\sigma$ represents the sigmoid activation function. $M^T$ denotes the learnable weight matrix.

Ultimately, the two hidden vectors of the last layer of both networks are concatenated as inputs for the neural matrix factorization layer. Both linear and non-linear features are integrated in the text-based collaborative filtering component. The interaction probability between the target dataset and the machine learning method is predicted as follows.

$$\hat{y}_2 = \sigma\left(W_t^T(\phi_l \parallel \phi_{nl})\right). \tag{22}$$

## Prediction

After incorporating the two critical components in FEGNN, the embedding representations for both the dataset and the method are obtained. This involved utilizing both the graph structure and text description information. As the two embedding representations are obtained through different learning methods, our approach enables the model to learn and optimize these representations independently. Subsequently, the final predicted probability is expressed as follows.

$$\hat{y}_{dm} = W^T\left(\sigma\left(\left(g_d^{(h)}\right)^T g_m^{(h)}\right) + \sigma\left(W_t^T(\phi_l \parallel \phi_{nl})\right)\right). \tag{23}$$

The loss function is defined as follows:

$$\mathcal{L} = \sum_{d \in D, m \in M}\left(\mathcal{J}\left(\hat{y}_1, y_{dm}\right) + \mathcal{J}\left(\hat{y}_2, y_{dm}\right)\right) + \lambda \parallel \Theta \parallel \tag{24}$$

where $\mathcal{J}$ is the cross-entropy function, when the actual output $\hat{y}_1$ or $\hat{y}_2$ is close to the desired output $y_{dm}$, the value of the cost function is close to zero. The cross-entropy function $\mathcal{J}$ helps to accelerate the speed of updating the weights. $\lambda$ controls the regularisation strength and $\Theta$ is the set of parameters.

## EXPERIMENTS

In this section, the performance of the proposed FEGNN method is evaluated by conducting experiments on the Machine Learning Dataset to address the following three questions.

1) How does the proposed FEGNN behave compared to the existing state-of-the-art baseline models?
2) Which part of FEGNN contributes to better performance?
3) What is the influence of the parameter settings on the effectiveness of the proposed FEGNN method?

**Table 1 The detailed messages about the Machine Learning Dataset.**

| Name | Amount |
|---|---|
| Machine learning dataset | 2,093 |
| Machine learning method | 7,644 |
| Types of machine learning task | 517 |
| Academic article | 4,338 |
| Open-source repository | 2,872 |

**Table 2 Statistics of data in the Machine Learning Dataset.**

| Knowledge graph | | Extracted dataset | |
|---|---|---|---|
| # Entities | 17,483 | # Datasets | 2,092 |
| # Relations | 23 | # Methods | 6,239 |
| # Triples | 117,245 | # Interactions | 13,732 |
| Avg.# descriptive words | 8.1 | # Density | 0.00105 |

## Dataset and preprocessing

The Machine Learning Dataset (*Cao et al., 2021*) utilized in this article includes machine learning-related datasets, methods, attributes, and relevant entities sourced from open academic platforms such as Paperswithcode and GitHub. Actually, 19 areas are involved in the dataset, such as computer vision, natural language processing, reinforcement learning, and so on. The detailed messages about the dataset are listed in Table 1. Descriptive information for the datasets and methods is extracted from the context of the tasks using the datasets, the titles or abstracts of articles related to the methods. Table 2 presents the fundamental statistics of the dataset.

## Baselines

This article evaluates the proposed FEGNN approach with various existing strong baselines.

- **BPR** (*Rendle et al., 2009*). This approach utilizes Bayesian analysis on the traditional factorization machine model for optimization.
- **KGCN** (*Wang et al., 2019b*). This approach utilizes the similarity of the relationship of different users to assign varying weights to neighbors to obtain a higher-order representation of the entity.
- **KGNN-LS** (*Wang et al., 2019c*). This approach generates personalized embeddings for each item by introducing label smoothing regularization based on KGCN.
- **KGAT** (*Wang et al., 2019a*). This approach employs the graph attention mechanism to evaluate the importance of different neighbors and derives entity representations for recommendation through graph neural networks.
- **CKE** (*Zhang et al., 2016*). This approach integrates collaborative filtering with structural, text, and visual knowledge into a unified recommendation framework.

- **MKGAT** (*Sun et al., 2020*). This approach performs the multimodal graph attention mechanism in multimodal graphs to obtain a better embedding representation.
- **DEKR** (*Cao et al., 2021*). This approach constructs a description-enhanced knowledge graph and combines knowledge graph-based and text-based approaches.
- **CKGC** (*Cao et al., 2022*). This approach achieves cross-modal knowledge graph contrastive learning by maximizing the agreement between the descriptive view representations and structural view representations.

### Experiments setup

**Evaluation Metrics:** In this experiment, Precision@K, Recall@K, and NDCG@K are used as Top-K recommendation metrics. Additionally, the evaluation metrics of click-through-rate (CTR) prediction are assessed through area under curve (AUC), Accuracy (ACC) and F1-score.

**Parameter settings:** The proposed FEGNN approach is implemented by using PyTorch. The dimensions for graph structure embedding and text embedding are set to 64, with each node having eight neighbors. Moreover, the number of graph convolution iterations is set to 2. The Adam optimizer (*Kingma & Ba, 2017*) is employed for all the methods to optimize the training process, and a batch size of 128 is chosen. A grid search is conducted for the learning rate and regularization factor, with values $\{10^{-4}, 5 \times 10^{-4}, \ldots, 10^{-1}, 5 \times 10^{-1}\}$ and $\{10^{-6}, 10^{-5}, \ldots, 10^{-2}, 10^{-1}\}$, respectively. The embedding size for all baseline models is set to 64. For KGCN, KGNN-LS and KGAT methods, the number of propagation hops is set to 2 and CKGC is set to 3. Specifically, for the KGCN method, the number of neighborhood samples is set to 8, and the sum aggregator is chosen as the aggregation method. For the KGAT method, the bi-interaction aggregator is employed for aggregation, and node drop is applied, just as KGAT suggested.

### Comparison of results (Q1)

Table 3 presents the Top-K recommendation results and Table 4 shows the results of CTR prediction, respectively. From these experimental results, the article makes the following observations:

- In most cases, FEGNN behaves the best both in the Top-K recommendation and CTR prediction. To validate the significance of the experimental results, the Unpaired Two Sample t-test (*Mishra et al., 2019*) with $p < 0.05$ is conducted on the CTR prediction results of DEKR and FEGNN. The significant results indicate that a notable improvement of over 40% in the Top-10 recommendation is observed, and over 55% in the Top-20 recommendation is captured by applying the proposed FEGNN method on the Machine Learning Dataset.
- Among the baseline methods, the hybrid recommendation models based on knowledge graphs, including MKGAT, DEKR, CKGC and FEGNN, achieve superior performances. In these hybrid recommendation models, the multimodal knowledge is believed to bring benefits for enhancing the recommendation performance. The performance of CKE also benefits from its consideration of multimodality, but the simple embedding of knowledge may hurt the effectiveness of the recommendation.

**Table 3 Overall comparison in Top-K recommendation.**

| | Top-10 recommendation | | | Top-20 recommendation | | |
|---|---|---|---|---|---|---|
| Models | Precision | Recall | NDCG | Precision | Recall | NDCG |
| BPR | 0.0360 (−158.33%) | 0.1132 (−243.29%) | 0.1107 (−147.88%) | 0.0205 (−256.10%) | 0.1552 (−250.84%) | 0.1227 (−162.18%) |
| KGCN | 0.0298 (−212.08%) | 0.0896 (−333.71%) | 0.0736 (−272.83%) | 0.0232 (−214.66%) | 0.1364 (−299.19%) | 0.0904 (−255.86%) |
| KGNN-LS | 0.0309 (−200.97%) | 0.0526 (−638.78%) | 0.0374 (−633.69%) | 0.0225 (−224.44%) | 0.0724 (−652.07%) | 0.0469 (−585.93%) |
| KGAT | 0.0591 (−57.36%) | 0.1431 (−171.56%) | 0.1209 (−126.96%) | 0.0373 (−95.71%) | 0.1932 (−181.83%) | 0.1381 (−132.95%) |
| CKE | 0.0511 (−82.00%) | 0.1178 (−229.88%) | 0.0995 (−175.78%) | 0.0342 (−113.45%) | 0.1438 (−278.65%) | 0.1135 (−183.44%) |
| MKGAT | 0.0615 (−51.22%) | 0.1786 (−117.58%) | 0.1317 (−108.35%) | 0.0416 (−75.48%) | 0.2601 (−109.34%) | 0.1672 (−92.40%) |
| DEKR | 0.0642 (−44.86%) | 0.2155 (−80.32%) | 0.1598 (−71.71%) | 0.0462 (−58.01%) | 0.3268 (−66.62%) | 0.1946 (−65.31%) |
| CKGC | 0.0860 (−8.14%) | 0.3603 (−7.85%) | 0.2527 (−8.59%) | 0.0715 (−2.10%) | 0.5267 (−3.38%) | 0.3097 (−3.87%) |
| FEGNN | 0.0875 | 0.3886 | 0.2744 | 0.0730 | 0.5445 | 0.3217 |

**Table 4 Overall comparison in click-through-rate prediction.**

| | CTR prediction | | |
|---|---|---|---|
| Models | AUC | Accuracy | F1-score |
| BPR | 0.7518 (−23.31%) | 0.6466 (−30.17%) | 0.6862 (−26.22%) |
| KGCN | 0.8112 (−17.37%) | 0.7366 (−21.17%) | 0.7435 (−20.49%) |
| KGNN-LS | 0.8231 (−16.18%) | 0.7221 (−22.62%) | 0.7506 (−19.78%) |
| KGAT | 0.8394 (−14.55%) | 0.7396 (−20.87%) | 0.7598 (−18.86%) |
| CKE | 0.7684 (−21.65%) | 0.6535 (−29.48%) | 0.6966 (−25.18%) |
| MKGAT | 0.8829 (−10.20%) | 0.7638 (−18.45%) | 0.7897 (−15.87%) |
| DEKR | 0.9687 (−1.62%) | 0.9172 (−3.11%) | 0.9197 (−2.87%) |
| CKGC | 0.9851 (+0.02%) | 0.9472 (−0.11%) | 0.9453 (−0.31%) |
| FEGNN | 0.9849 | 0.9483 | 0.9484 |

• Both CKGC and FEGNN perform significantly better than DEKR. DEKR obtains a comprehensive representation of entities by combining knowledge graph-based and text-based approaches. CKGC achieves contrastive knowledge learning by maximizing the agreement between the descriptive view representations and structural view representations. FEGNN captures a more comprehensive graph structure representations through the feature-enhanced graph neural network.

• Compared with CKGC, FEGNN behaves better when Top-K recommendation is taken as the evaluation measure. The click-through-rate prediction by applying both CKGC and FEGNN, are comparable. It is suggested that the graph structure representation obtained through the traditional graph neural network limits the recommendation performance of CKGC, while more comprehensive graph structure representations are captured in FEGNN through a novel feature-enhanced graph neural network.

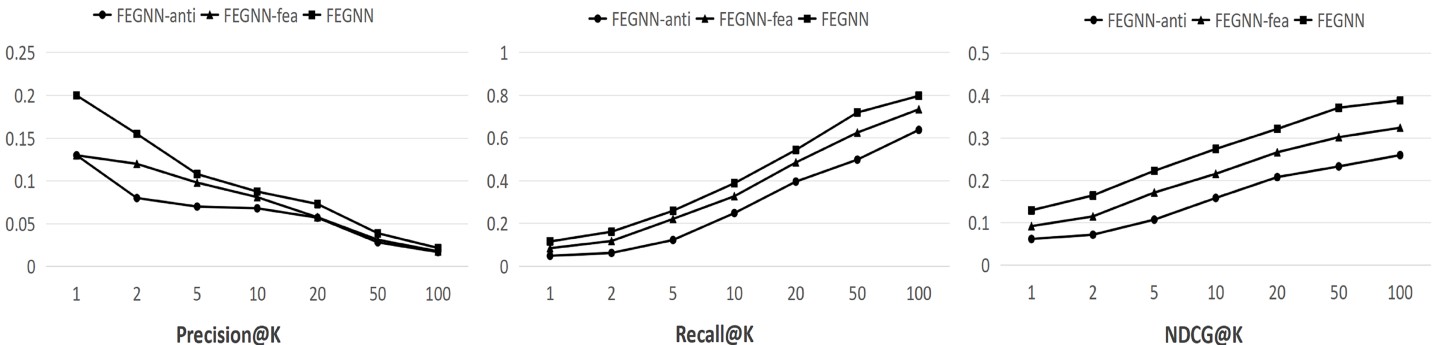

**Figure 2 The impact of anti-smoothing networks and feature enhancement on model performance.**

| Table 5 Ablation experimental results. | | | |
| --- | --- | --- | --- |
| Models | AUC | Accuracy | F1-score |
| FEGNN-anti | 0.9684 | 0.9139 | 0.9141 |
| FEGNN-fea | 0.9805 | 0.9374 | 0.9372 |
| FEGNN | 0.9849 | 0.9483 | 0.9484 |

## Ablation experiment (Q2)

To verify whether node feature enhancement and the anti-smoothing aggregation network can improve the performance of recommendation, ablation experiments are conducted. Two variants of FEGNN are devised: FEGNN-anti (only the anti-smoothing network contained) and FEGNN-fea (only feature enhancement contained).

Based on the results in Fig. 2 and Table 5, it is evident that FEGNN-fea brings notable advantages in improving the recommendation performance while FEGNN-anti has minimal impact on the recommendation performance. It is attributed to the limited number of model iterations and neighborhood samples, making them less susceptible to embedding smoothing. Nevertheless, as the propagation aggregation process relies on the feature-enhanced nodes, the entity embedding contains more information and is more susceptible to embedding smoothing issues. Therefore, the effectiveness of the proposed FEGNN method is further improved by combining feature enhancement and the anti-smoothing network.

## Study of parameters (Q3)

This section examines the effects of different numbers of sampled neighbors, propagation layers, and embedding dimensions.

**Impact of neighbor sampling size:** We assess the model's performance by varying the number of neighbor samples. Table 6 indicates that the optimal results can be fetched when the number of neighbor samples is set to eight or 16. It is believed that each entity and relationship is relatively condensed in the machine learning knowledge graph

**Table 6 Experiments results with different number of neighborhood samples.**

| K | AUC | ACC | F1-score |
|---|---|---|---|
| 2 | 0.9792 | 0.9321 | 0.9328 |
| 4 | 0.9802 | 0.9335 | 0.9340 |
| 8 | 0.9849 | 0.9483 | 0.9484 |
| 16 | 0.9824 | 0.9434 | 0.9434 |
| 32 | 0.9805 | 0.9402 | 0.9393 |

compared to other knowledge graphs. Specifically, a too-small value for $K$ lacks the capacity to collect sufficient neighborhood information, while a huge $K$ introduces noise, which may lead to a decline in recommendation performance. $K$ is set to eight in the experiments to strike a balance between effectiveness and efficiency.

**Impact of receptive field depth:** The number of iterations $H$ is varied from 1 to 4. The results in Table 7 indicate that the optimal performance occurs with two and three iterations. However, when the iteration is set to 4, the recommendation performance declines significantly, accompanied by a substantial increase in training time. This decline is attributed to insufficient graph structure information. In the process of inter-item similarity inference, excessively long connectivity relationships between entities are almost meaningless (*Wang et al., 2019b*). Based on these experimental results, the number of iterations is set to 2 in the experiments.

**Impact of embedding dimension:** The dimensions of entity embedding are varied to analyze experimental results in Table 8. Initially, the recommendation performance is enhanced by increasing $d$. However, an excessively high value for $d$ becomes sensitive, which negatively impacts the recommendation performance.

## DISCUSSION

Figure 3 plots the trends of AUC curve with different numbers of iteration by utilizing FEGNN, CKGC and DEKR. It is obvious that the best results of CKGC are obtained when the iteration number is set to 3. For FEGNN and DEKR, the best recommendation effectiveness is achieved when the iteration number is 2. In GNN, expensive computations in batched training and inference are caused by the recursive expansion of neighborhoods across layers, which indicates that when the layers of GNN increase, the time cost of inference increases almost exponentially (*Yan et al., 2020*). The same components of GNN are adopted to learn the graph structure information in the three methods, thus the method with fewer iterations is believed to consume less time and computations. Therefore, FEGNN achieves a comparable recommendation effectiveness with less time when compared with CKGC. Although the trend of AUC curve for DEKR is consistent with that of FEGNN, FEGNN behaves significantly better than DEKR in Top-K recommendation and CTR prediction.

In addition, CKGC uses extra loss functions for contrastive learning and more parameter settings, which is another time-consuming process. Therefore, compared with

**Table 7 Experiments results with different number of iterations.**

| H | AUC | ACC | F1-score |
|---|-----|-----|----------|
| 1 | 0.9608 | 0.9057 | 0.9070 |
| 2 | 0.9849 | 0.9483 | 0.9484 |
| 3 | 0.9826 | 0.9397 | 0.9404 |
| 4 | 0.9791 | 0.9383 | 0.9382 |

**Table 8 Experiments results of different dimension sizes.**

| d | AUC | ACC | F1-score |
|---|-----|-----|----------|
| 4 | 0.8847 | 0.8022 | 0.8160 |
| 8 | 0.9460 | 0.8726 | 0.8779 |
| 16 | 0.9660 | 0.9015 | 0.9046 |
| 32 | 0.9756 | 0.9219 | 0.9227 |
| 64 | 0.9849 | 0.9483 | 0.9484 |
| 128 | 0.9797 | 0.9403 | 0.9406 |

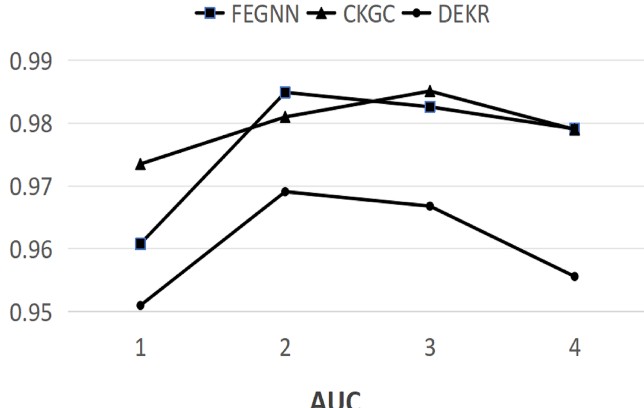

**Figure 3 AUC obtained by applying different methods with different number of iterations.**

CKGC, FEGNN achieves comparable recommendation effectiveness with less space and time in CTR prediction, while achieving notable advantages in Top-K recommendation.

## CONCLUSION AND FUTURE WORK

This article proposes a knowledge graph-based recommendation framework that integrates two novel components: a feature-enhanced graph neural network and an anti-smoothing aggregation network. The rich entity connections in the text-enhanced machine learning knowledge graph are leveraged to recommend suitable machine learning methods for a given dataset. To mind the rich latent representation for each entity, the representation of nodes involved in the propagation process of the target entity is enhanced by aggregating the direct neighbor information. Also, an anti-smoothing

aggregation network is designed to reduce the impact of over-smoothing with the increase in the number of iterations.

In the future, it is worth striking a balance between utilizing auxiliary information in the knowledge graph and avoiding over-smoothing issues, which is an area of exploration. In addition, considering the diverse types of tasks in machine learning, such as clustering, classifying, and so on, recommending more relevant machine learning methods based on the class of the machine learning task is a challenge.

### Funding
This work is supported by the University Natural Sciences Research Project of Anhui Province (KJ2021A0993) and the Program for Scientific Research Innovation Team in Colleges and Universities of Anhui Province (2022AH010095). The funders had no role in study design, data collection and analysis, decision to publish, or preparation of the manuscript.

### Grant Disclosures
The following grant information was disclosed by the authors:
University Natural Sciences Research Project of Anhui Province: KJ2021A0993.
Scientific Research Innovation Team in Colleges and Universities of Anhui Province: 2022AH010095.

### Competing Interests
The authors declare that they have no competing interests.

### Author Contributions
- Xin Zhang conceived and designed the experiments, performed the experiments, analyzed the data, performed the computation work, prepared figures and/or tables, authored or reviewed drafts of the article, and approved the final draft.
- Junjie Guo conceived and designed the experiments, performed the experiments, analyzed the data, performed the computation work, prepared figures and/or tables, authored or reviewed drafts of the article, and approved the final draft.

### Data Availability
The Machine Learning Dataset is available at GitHub and Zenodo:
- https://github.com/cxsss/DEKR/tree/main/DEKR/data/MachineLearning
Please use the command statement "python main.py" to run the program.
- Xianshuai Cao, Yuliang Shi, Han Yu, Jihu Wang, Xinjun Wang, Zhongmin Yan, & Zhiyong Chen. (2024). MachineLearningDataset [Data set]. Zenodo. https://doi.org/10.5281/zenodo.13328888.

## Supplemental Information

Supplemental information for this article can be found online at http://dx.doi.org/10.7717/peerj-cs.2284#supplemental-information.

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
