# Peer review of "A feature-enhanced knowledge graph neural network for machine learning method recommendation"

_PeerJ Computer Science, doi:10.7717/peerj-cs.2284_

## Round 0.1 · original submission · Major Revisions

Please revise the work according to the comments. Then it will be evaluated again.

Reviewer 1 ·

Basic reporting

This paper proposes a recommendation framework that integrates knowledge graph and text-based methods to address the challenges faced when selecting suitable machine learning methods for target datasets. In the proposed framework, to obtain richer latent representations, each node is represented not only by descriptive information in the knowledge graph but also by involving its neighborhood information to enhance the representation, capturing more comprehensive higher-order information during entity propagation. I find the ideas presented in the paper interesting and meaningful, but I still have some concerns as follows:
1. The model framework FEGCN proposed in the paper seems to share many similarities with the state-of-the-art baseline model (DEKR), such as using linking, graph multi-hop propagation, and fusion of two embedding representations. Therefore, I believe this paper should be positioned as a modification of the DEKR model. If the authors have taken this approach, it needs to be clearly stated in the paper how it differs from DEKR.
2. There are still some Chinese characters in Figure 2 of the paper, and some symbols are unclear due to deformation.

Experimental design

1. I believe it would be beneficial to test the proposed model on different datasets because the improvement over the DEKR model is not significant, and testing on a single dataset may introduce randomness.
2. The paper could provide an introduction to the formulas of the evaluation functions.

Validity of the findings

As it stands, the paper lacks innovation. I hope the authors can clearly explain the differences from DERK in the paper and highlight the specific contributions of this paper compared to DERK.

Additional comments

no comment

Reviewer 2 ·

Basic reporting

1. Enhance English writing skills. For example, the phrase "The first-order neighbours of 'MovieLens-20M' are 'Tasks' such as CTR Prediction, Recommender Systems, Knowledge Base. Besides 'Methods' such as KNI, Deep-FM, and KGCN are the first-order neighbours" is confusing. Additionally, the causal relationship in "However, in most knowledge graph-based recommendation methods, each entity is only represented by its name and informative messages carried by are ignored. Therefore, the paper captures the enhancement features for each node by aggregating their first-order neighbour information" is weak.
2. The Introduction section dedicates a significant amount of space to discussing Figure 1. However, we advise to focus on categorizing or discussing the development process of research papers in the field.
3. The research direction of the paper is knowledge graph + graph networks, but there is a lack of introduction to graph networks in the related work section. It is recommended to include relevant works.
4.The piecewise function in Equation 1 needs to be aligned properly.
5.Normalize the notation of the equations. For example, in Equation 2, it is recommended to use different symbols 'e' for the numerator and denominator, and avoid using the same symbol '\phi' for both the input and output before and after normalization.
6.Symbols are missing on lines 207-208.

Experimental design

This study is relatively novel. However, the latest baseline algorithm used is from 2021. More recent baseline algorithms within the past three years should be included.

Validity of the findings

no comment

---

## Round 0.2 · Minor Revisions

The authors addressed some issues. However, the experiments are not enough. Moreover, new references should be analyzed.

Reviewer 1 ·

Basic reporting

no comment

Experimental design

no comment

Validity of the findings

no comment

Reviewer 2 ·

Basic reporting

N/A

Experimental design

N/A

Validity of the findings

N/A

Additional comments

1. In the revised manuscript, the latest baseline used for comparison is the CKGC algorithm in 2022. Furthermore, even compared with CKGC, the proposed FEGNN method obtains trivial advantage, particularly in CTR Prediction. It is crucial to identify the optimization improvements of FEGNN in terms of time complexity, space complexity, or other relevant factors to substantiate its significant superiority over the baseline.
2. The manuscript exhibits a comparatively low proportion of references from the past three years, and the Introduction section fails to adequately review the developments and achievements in the research field. This indicates a lack of systematic and in-depth exploration of the domain in the manuscript. Further modifications are required to address these concerns.

---

## Round 0.3 · accepted · Accept

Thanks to the authors for your efforts to improve the article. I'm pleased to inform you that this version is ready for acceptance. Congrats!

Reviewer 2 ·

Basic reporting

N/A

Experimental design

N/A

Validity of the findings

N/A

Additional comments

This revised version has addressed all of my concerns, and I believe this paper can now be accepted.